# Animal Models for Influenza Research: Strengths and Weaknesses

**DOI:** 10.3390/v13061011

**Published:** 2021-05-28

**Authors:** Thi-Quyen Nguyen, Rare Rollon, Young-Ki Choi

**Affiliations:** 1College of Medicine and Medical Research Institute, Chungbuk National University, Cheongju 28644, Korea; quyenbio@gmail.com (T.-Q.N.); rarerollon@gmail.com (R.R.); 2Zoonotic Infectious Diseases Research Center, Chungbuk National University, Cheongju 28644, Korea

**Keywords:** influenza, pathogenesis, transmission, vaccine efficacy, antiviral drug, mice, ferrets, guinea pigs, non-human primates

## Abstract

Influenza remains one of the most significant public health threats due to its ability to cause high morbidity and mortality worldwide. Although understanding of influenza viruses has greatly increased in recent years, shortcomings remain. Additionally, the continuous mutation of influenza viruses through genetic reassortment and selection of variants that escape host immune responses can render current influenza vaccines ineffective at controlling seasonal epidemics and potential pandemics. Thus, there is a knowledge gap in the understanding of influenza viruses and a corresponding need to develop novel universal vaccines and therapeutic treatments. Investigation of viral pathogenesis, transmission mechanisms, and efficacy of influenza vaccine candidates requires animal models that can recapitulate the disease. Furthermore, the choice of animal model for each research question is crucial in order for researchers to acquire a better knowledge of influenza viruses. Herein, we reviewed the advantages and limitations of each animal model—including mice, ferrets, guinea pigs, swine, felines, canines, and non-human primates—for elucidating influenza viral pathogenesis and transmission and for evaluating therapeutic agents and vaccine efficacy.

## 1. Introduction

Influenza viruses are enveloped RNA viruses belonging to the *Orthomyxoviridae* family [1] and are classified as A, B, C, or D based on antigenic differences [2]. Influenza virus infection is a global public health concern due to the risk of pandemics with high levels of mortality and morbidity worldwide [3]. The potential for severity is increased by the fact that influenza can transmit with high efficiency to the human population from animal reservoirs. These viruses frequently undergo genetic reassortment, including antigenic shift and drift, to generate new viruses, occasionally giving rise to unpredictable pandemics. Previous influenza pandemics include the 1918 Spanish H1N1, 1957 Asia H2N2, and 1968 Hong Kong H3N2 pandemics, as well as the 2009 H1N1 outbreak, which resulted in an estimated 300,000 deaths within the first 12 months—making it the fourth greatest pandemic of the last 100 years [3,4]. In addition, seasonal influenza epidemics cause up to 650,000 deaths and massive economic burdens every year [5]. 

Even though numerous studies have been performed in the over 100 years since the 1918 influenza pandemic, knowledge of the host factors influencing influenza disease severity remains elusive. Shortcomings include understanding the transmission mechanisms, natural history and precise pathogenesis of influenza disease, and host immune responses. In addition, knowledge gaps exist regarding the relationship between clinical presentation, transmission, and protection levels. Given that universal influenza vaccines are still unavailable, there remains prodigious potential for influenza to reassort and cause severe human epidemics and pandemics. Therefore, it is essential to continuously assess host-virus interactions, transmission mechanisms, and the host immune response to different influenza viruses in various animal models. The selection of appropriate animal models for specific research questions is prerequisite for accurate understanding of influenza virus properties prior to clinical trials for novel universal influenza vaccines. In this review, the advantages and disadvantages of different animal models used for influenza research, including mice, ferrets, guinea pigs, swine, felines, canines, and non-human primates, will be discussed.

## 2. Mouse Model

Mice are the experimental tool of choice for study of infectious diseases for several reasons: (1) easily manipulated genome, (2) rapid reproductive rate and ease of handling, (3) ease of husbandry, (4) low cost, including animals and housing, and (5) decreased variability.

In influenza research, mice are the most widely used animal model, and among inbred mice, C57BL/6 and BALB/c are the most commonly used. Influenza-infected mice develop clinical symptoms, such as anorexia, malaise, and cytokine storm, depending on mouse strain, virus strain, and challenge dose [1]. As in humans, reduced blood oxygen saturation, increased lactate dehydrogenase, and total protein levels can be used to measure pneumonia in mice. Bodyweight loss and survival rate of mice are also good markers of influenza disease severity. However, studies have shown that the pathogenesis of the influenza virus in mice does differ to some extent from that in humans. Unlike humans, mice infected with the influenza virus do not show fever, instead exhibiting hypothermia [6]. Some clinical signs, such as cyanosis, dyspnea, and hemoptysis, are also not as easily observed in mice [7]. Studies demonstrated that the use of DBA/2 mice enhanced influenza virus pathogenesis, resulting in significant body weight loss, higher mortality, enhanced cytokine production, and more severe lung pathology compared to C57BL/6 mice [8,9]. Other mouse models, such as Tmprss2^−/−^ mice, have also been used to study the critical role of host protease TMPRSS2 in influenza viral pathogenesis [10,11,12,13].

Compared to other animal models, there appears to be limited utility for mouse models in the study of influenza virus transmission [14]. Although Schumal et al. successfully developed a transmission mouse model in the 1960s, it was found that only H2N2 subtypes (meaning no other subtypes, including H1N1, H3N2, or H5N2) were transmissible in their system [15,16,17,18,19]. Thus, transmission of influenza in mice is only possible under certain conditions, or with specific mouse strains and mouse-adapted virus isolates. For example, Edenboroughand et al. demonstrated that direct contact is an important factor for virus transmission in mice [20]. However, Ortigoza and colleagues recently showed that an infant C57BL/6 mouse model might be useful for studying influenza virus transmission [21], and DRAGA mice have also been recently used as a transmissible model for influenza A virus infections [22]. Thus, further studies should be conducted to better understand influenza transmission in a mouse model.

Although the pathogenesis of the influenza virus is not completely recapitulated in mice, the convenience of monitoring clinical signs, including bodyweight loss, has resulted in their use for investigation of influenza vaccines and antiviral drug efficacy in many preclinical studies. Furthermore, the availability of commercial reagents specific for mice facilitates the investigation of the immune response against influenza virus infection. For instance, the availability of cytokine and chemokine detection kits helps scientists to observe the influenza-induced cytokine storm, which is also seen in humans, and to investigate Th1, Th2, and Th17 cytokine responses after immunization. The use of mouse-specific antibodies, together with flow cytometric analysis, enzyme-linked immunosorbent assay (ELISA), or other relevant techniques, allows measurement of immune cell responses and antigen-specific antibody responses after vaccination. In addition, researchers have used antibodies to deplete immune cells, such as CD4, CD8, or NK cells, to study the contribution of these cells to the protection against influenza infection in vaccinated mice [23,24]. Preclinical studies using knockout or deficient mouse models have also elucidated the roles of immune cells, including B cells, T cells, NK cells, and monocytes [25,26,27,28,29,30,31,32,33,34], as well as the IFN signaling pathway [35,36] (Table 1). Transgenic mice, such as B6-Mx1^−/−^ carrying mutant *Mx1* alleles, B6-Mx1^r/r^ carrying a functional A2G *Mx1* allele, and SPRET/Ei, which carry another *Mx1* wild-type allele, have been applied to study the importance of the *MX1* gene in influenza virus resistance [37,38]. DRAGA mice that lack the murine immune system and express a functional human immune system were recently used to generate cross-reactive, anti-human influenza monoclonal antibodies [39]. The employment of humanized mice allows studies of influenza vaccine and antiviral drug efficacy and safety [40,41,42,43,44] (Table 1). For example, HLA-A2 mice expressing a hybrid HLA-A2 transgene comprising the human α1/α2 domains and β2-microglobulin fused with a murine α3 domain were used to investigate the effects of influenza polypeptide vaccines [41]. Nonobese, diabetic, severe combined immunodeficiency β_2_ microglobulin^−/−^ (NOD/SCID β2m^−/−^) mice (engrafted with human CD34^+^ hematopoietic progenitors and further reconstituted with T cells), can be used to test influenza vaccines that cannot be tested in human volunteers [42]. The NOD/SCID/Jak3^−/−^ (NOJ) mouse strain, which has defective T and B cells and impaired NK cell development, were transplanted with human peripheral blood mononuclear cells (huPBMC) to recapitulate the low immunogenicity of H7N9 influenza vaccines [43]. Rag2^−/−^γc^−/−^ mice reconstituted with huPBMC were used to investigate the effect of antiviral aminobisphosphonate pamidronate [40]. NOD/Shi-scid *IL2rγ^null^* mice, retaining human CD14^+^ cells, plasmacytoid dendritic cells, CD4^+^ and CD8^+^ T cells, and B cells in the lungs were also demonstrated to be suitable for evaluating the safety of influenza vaccines [44]. Kenney et al. applied IFITM3^−/−^ mice, in which IFITM3 genes were deleted by CRISPR/Cas9-based deletion strategy, to evaluate influenza-induced cardiac pathogenesis [45].

In spite of the significant advantages, there are concerns over the use of mouse models for biomedical research in general. These include the differences between murine and human immune systems, such as frequencies of lymphocytes and neutrophils in peripheral blood, and differences in toll-like receptors, defensins, and IgG classes [46]. With the exception of high pathogenetic strains, such as pandemic H1N1 2009 [47], H5N1 [48], H7N7 [49], and H7N9 [50], mice are not naturally infected with influenza viruses. Therefore, viruses must be adapted through multiple lung passages to allow for infection and replication. Moreover, because influenza viruses do not transmit well among mice, more suitable animal models must be used to study this aspect [1,51]. In addition, unlike humans, mice exhibit influenza A virus replication in the lower respiratory tract; thus, IAV virulence is not preserved between these species [52]. 

## 3. Ferret Model

Ferrets have been used as an animal model for influenza research since 1933 when Smith et al. successfully isolated the human influenza A virus via inoculation into ferrets [53]. Recently, the successful genome sequencing of ferrets [54] has facilitated the use of this mammalian model in infectious disease research. Thus, ferrets have been used in a wide array of influenza research, including assessment of virus pathogenicity, transmissibility, viral tropism, host immune responses, and evaluation of novel vaccines and antiviral treatments [55] (Table 2).

Ferrets and humans share similar lung pathology and cellular receptor distribution [56], making ferrets attractive models for studies of the pathogenicity, transmissibility, and tropism of influenza viruses [57]. In addition, ferrets are naturally and highly susceptible to many different influenza virus strains, including influenza A and B viruses [58]. Previous studies showed that ferrets also exhibit predominant infection of influenza virus in the upper respiratory tract, as seen in humans. In addition, highly virulent influenza viruses can also infect the lower respiratory tract of ferrets [57]. Furthermore, influenza-infected ferrets exhibit similar clinical symptoms as seen in humans, such as fever, nasal secretions, coughing, gastrointestinal complications, serum abnormalities, neurological complications, weight loss and/or anorexia, lymphopenia, hypercytokinemia, and lethargy [14,57,58,59,60]. 

Human-to-human transmission of influenza viruses remains poorly understood due to knowledge gaps in the relationship between clinical symptoms, virus shedding, and transmission [61]. Unlike mice, influenza-infected ferrets are capable of transmitting the virus through both direct and indirect contact (aerosol, respiratory droplet, or airborne transmission) and thus, can be used as models to study influenza virus transmission [14] (Table 2). The assessment of transmission is performed by detecting the virus in nasal secretions and investigating seroconversion [62]. Zhou and colleagues used this model to demonstrate that the size of airborne particles dictated the transmission through the air of 2009 pandemic H1N1, recombinant human seasonal H3N2, and triple-reassortant swine H1N2 influenza viruses in ferrets [63]. A previous study used ferrets to assess virus transmission by direct contact of the highly pathogenic avian H5N2 and H5N6 viruses, which cause lethal diseases in humans. The study showed that the transmission between co-housed ferrets was limited to H5N6, but not the H5N2 virus [64]. Herfst et al. reported that the H5N6 virus causes severe disease, but does not transmit between ferrets via the airborne route [65]. In addition, R195K mutation in the PA-X protein was found to increase the virulence and transmission of pandemic H1N1 influenza A viruses via direct contact and aerosol transmission in the ferret model [66]. The impact of HA stability on the transmission of swine H1N1 gamma influenza virus (airborne transmission) [67] and avian H10N7 influenza virus (aerosol or respiratory droplet transmission) [68] was also demonstrated using ferret models. It should be noted that although ferrets are currently used as models for influenza transmission studies, there is a debate about the potential influenza pandemics associated with studies on transmission of highly pathogenic influenza viruses using ferrets [69,70,71,72].

**Table 2 viruses-13-01011-t002:** Summary of some discoveries in influenza virus research using the ferret model in 2020–2021.

Research Area	Research Discovery	Year	Reference
Pathogenesis and transmission	The importance of pre-existing heterosubtypic immunity to airborne transmission of influenza viruses	2021	[73]
	Effect of posttranslational modifications such as SUMOylation on the adaptation, pathogenesis, and transmission of IAVs	2021	[74]
	The wild birds-derived H9N2 virus exhibits efficient transmissibility in mammalian models via respiratory droplets	2021	[75]
	The matrix gene of the pandemic H1N1 virus contributes to the pathogenesis and transmission of the swine influenza virus	2021	[76]
	The role of HA pH of fusion on the transmissibility of a cell culture-adapted H3N2 virus	2021	[77]
	H3N2 virus isolated from swine replicates in ferrets and transmits from swine to ferret	2020	[78]
	Effects of influenza haemagglutinin stability on influenza virus transmission	2020	[67,68,79]
	R195K mutation in the PA-X protein increases the virulence and transmission of IAVs	2020	[66]
	Influenza A viruses are transmissible via the air from the nasal respiratory epithelium	2020	[80]
Vaccine and antiviral treatments	H2HA vaccine elicits cross-reactive antibodies in influenza virus preimmune ferret models	2021	[81]
	H7N9 inactivated split virion vaccines adjuvanted with AS03 induces cross-reactive antibody responses and provided protection against H7N9 virus	2021	[82]
	Inactivated pandemic 2009 H1N1 IAV vaccine induces different protective efficacy following homologous challenge	2021	[83]
	Chimeric HA–based live attenuated vaccine provides long-term immunity against IAV	2021	[84]
	Low viral fitness leading to interstrain competition is the root cause of reduced H1N1 live-attenuated vaccine effectiveness	2021	[85]
	H7N9 split influenza vaccine adjuvanted with SWE adjuvant enhances antibody response and protection against severe pneumonia	2020	[86]
	MDCK-based H5 and H7 vaccines are comparable to the egg-based live attenuated vaccine in immunogenicity	2020	[87]
	Vaccination of adeno-associated virus-vectored vaccine reduces influenza disease severity	2020	[88]
	Seasonal H1N1 influenza vaccine induces systemic and respiratory T cell response conferring protection against H2N2 virus	2020	[89]
	DNA vaccine protects against the homologous H1N1 virus challenge	2020	[90]
	The combination of nanoemulsion and CpG enhances the effective immune response against IAV	2020	[91]
	Treatment of Bolozavir reduces onward transmission of pandemic H1N1 virus-infected ferrets	2020	[92]
	The risk of transmission of Baloxavir drug-resistance viruses from treated ferrets	2020	[93]
	Influenza clinical drug candidate EIDD-2801 reduces viral shedding and increased humoral responses to IAVs	2020	[94]
	The treatment of human plasma-derived IgG product (FLU-IGIV) reduces viral load in lungs of pandemic H1N1-infected ferrets	2020	[95]

Ferrets are also an indispensable animal model for evaluating the host immune response to influenza as well as antiviral drug and vaccine efficacies (Table 2). Early studies using the ferret model showed that cellular immune responses, especially cytotoxicity T cells, play an important role in recovery from influenza virus infection [96,97]. Furthermore, several influenza vaccine platforms have been studied using ferret models [81,82,83,86,87,98,99,100]. A study by Holzer et al. described differential protective efficacy of the Signal Minus FLU vaccine candidate in ferrets and pigs [101]. In addition, researchers demonstrated that vaccination of ferrets with chimeric hemagglutinin-based live attenuated vaccine provided durable protection against influenza A viruses [84]. Cross-protection of different influenza vaccines [99,100,102] and the role of adjuvants in influenza vaccines [82,86,91,99] have also been studied using ferret models. Additionally, many previous studies used ferrets to assess the effects of antiviral drugs against influenza virus infection [92,93,94,95,103]. Beale et al. employed ferrets to demonstrate that metabolomics could be used as indicators for the early diagnosis of influenza infection and to assess the effectiveness of drug therapies [103]. Treatment with human plasma-derived IgG product (FLU-IGIV) reduced viral loads in the lungs of pandemic H1N1-infected ferrets [95]. Lee et al. showed that Baloxavir, a recently licensed antiviral drug for influenza, could reduce onward transmission of pandemic H1N1 virus in infected ferrets [92]. However, it should be noted that Jones et al. reported that there is a risk of transmission of Baloxavir drug-resistant viruses from treated ferrets [93].

A limitation of the ferret model is the lack of reagents, including ferret-specific monoclonal antibodies, for analysis of immune responses to influenza virus infection and vaccination. Other disadvantages of ferrets as an experimental model are their size, high cost, and husbandry requirements, which make this model inaccessible to some researchers. Thus, due to the use of limited numbers of ferrets used in experiments, there is a possibility of misinterpretation of the results, which requires researchers trained well in doing experiments using this model. Of note, there is the possibility of danger in handling highly pathogenic influenza viruses in this model due to the risk of transmission. Experiments should be carefully performed with appropriate equipment in high-level biological safety laboratories.

## 4. Guinea Pig Model

Since 1963, guinea pigs (*Cavia porcellus*) have been occasionally used to study pathogenesis of the influenza virus [104,105,106]. Of note, the lung anatomy and physiology of the guinea pig resemble that of humans [107]. In addition, the commercial availability, small size, ease of handling and housing, and relatively low cost strengthen the use of this model for study of influenza viruses. 

Guinea pigs are naturally susceptible to various influenza virus strains without prior adaptation. Following infection of guinea pigs, the influenza virus mainly replicates in the upper respiratory tract; however, signs of disease are not regularly observed [14]. Further, influenza virus infection does not cause mortality in this model [108]. 

Together with ferrets, guinea pigs have been widely used to evaluate the transmission potential of influenza viruses. Lina et al. employed guinea pigs to show that adaptive amino acid substitutions account for the increased transmissibility of H9N2 avian influenza virus [109]. Mutations in the HA and PB1 genes were also demonstrated to increase pathogenicity and transmissibility of influenza viruses in the guinea pig model [110,111]. Previous studies using guinea pigs have shown that IAVs are transmissible via aerosolized fomites, microscopic particulates released from virus-contaminated surfaces (such as fur, skin, or bedding) [112], or respiratory droplets [113]. More recently, Asadi et al. described some unknown fraction of transmission events between guinea pigs in which aerosolized fomites were thought to be more important than respiratory droplets [114]. Wild bird-origin H5N6 avian influenza virus was also reported to be transmissible in guinea pigs [115].

Numerous influenza vaccine studies have been conducted in guinea pigs. McMahon et al. reported the role of neuraminidase-specific mucosal immunity in preventing influenza B virus transmission in guinea pigs [116]. In addition, vaccination with a live attenuated influenza vaccine prevented highly pathogenic H7N9 virus replication and transmission in guinea pigs [117]. Moreover, a consensus influenza D virus hemagglutinin-esterase fusion DNA vaccine completely protected guinea pigs from challenge with influenza D virus [118]. In addition, the effects of a plant-produced influenza human monoclonal antibody on preventing influenza infection and transmission in guinea pigs [119] suggests an alternative approach for development of immunotherapeutics for influenza treatment.

It should be noted that study of influenza viruses using guinea pigs has limitations. Guinea pigs show very few clinical symptoms following influenza infection, making it difficult to study viral pathogenesis. In addition, the lack of reagents specific to this model is also a drawback.

## 5. Hamster Model

The Syrian hamster (*Mesocricetus auratus*) is another of the animal models used for influenza virus pathogenesis, transmission, and vaccine efficacy studies, due to its ease in handling, reproduction capability, and relatively inexpensive maintenance cost [51]. This model was established due to its susceptibility to human influenza (H5N1, H9, and H3N2) and also to various avian influenza viruses without prior adaptation [51,120]. Replication and pathogenicity of several H9 isolates were investigated in this animal model [120]. Iwatsuki-Horimoto and colleagues [121] reported that Syrian hamsters were susceptible to H3N2 influenza viruses. Moreover, airborne transmission of pandemic H1N1 and seasonal H3N2 influenza virus isolates were also reported using Syrian hamsters. Heath et al. utilized hamsters and hamster tracheal organ cultures for virulence study of recombinant influenza viruses [122]. The dissemination of ingested H5N1 influenza viruses was also explored using the hamster model [123]. Results revealed that H5N1 can systemically spread through the hematogenous route, causing interstitial lung lesions.

Hamsters are also a crucial animal model for investigating influenza vaccine efficacy. Similar to humans, hamsters have a body temperature of 37 °C; thus, this animal model is suitable to study cold-adapted influenza virus vaccine efficacy [51]. Using hamsters as an experimental animal model, Potter et al. showed that vaccination with inactivated H3N2 influenza vaccine provided protection against the influenza virus challenge [124]. The enhanced antibody responses to inactivated influenza virus vaccine were also demonstrated in this model [125].

It should be noted that hamsters have not been extensively used in influenza studies given the disadvantage of the absence of clinical signs even with high viral titers in the respiratory tract. In addition, limited availability of reagents for immunological assays pose a disadvantage to attempts to further elucidate immunological responses in this animal model.

## 6. Chicken Model

The chicken (*Gallus gallus domesticus*) is also an ideal animal model for influenza research due to the high abundance of α2, 3 receptors present in the lower respiratory tract. This species has the advantage of low cost, high availability, with medium availability of species-specific reagents. It can also be genetically modified [126]. The use of the chicken model has provided insights into the understanding of influenza virus pathogenesis. A study reported that chickens infected with influenza virus typically produce signs of coughing, nasal discharge, respiratory distress, diarrhea, lethargy, and death [126]. Using chicken models, researchers have found that the high pathogenicity in H5N1 influenza viruses was caused by a wider tissue tropism of the viruses. Ranaware et al. utilized chickens to study host genetic components and their responses to influenza virus infection [127].

Chickens have also been used to investigate the efficacy of influenza vaccines. A study by Steel et al. reported that vaccination with a live attenuated vaccine containing non-structural protein 1 completely protected chickens from challenges of homologous high pathogenic avian influenza H5N1 virus (A/Viet Nam/1203/04) and showed a high level of protection against heterologous H5N1 virus (A/egret/Egypt/01/06) [128]. Van der Goot and colleagues investigated the effects of vaccine candidates on H7N7 influenza virus transmission using chicken models [129]. The use of T-cell-suppressed chickens demonstrated the role of T cell responses in influenza virus clearance after challenge with lowly pathogenic H9N2 virus [130].

Although the chicken model has been used for a wide array of influenza virus studies, this animal model is not susceptible to human influenza virus strains. Thus, chickens are not the experimental tool of choice for studying human influenza viruses.

## 7. Swine Model

Pigs *(Sus scrofa*) are a natural host of influenza A viruses. The abundance of α2, 6-linked sialic acid in the upper respiratory tract and α2,3-linked sialic acid in the lower respiratory tract of pigs makes them ideal hosts and “mixing vessels” for both avian and human influenza virus infections and reassortment events [108]. It was previously reported that swine and humans share marked similarities in genome sequences, anatomy, and physiology [131]. Therefore, the pig is an animal model of choice for studying influenza pathogenesis and transmission and for evaluating influenza vaccines.

A study by Barnard reported that pigs develop signs of diseases, including fever, coughing, loss of appetite, and labored breathing, after influenza virus infection [7]. Moreover, swine are susceptible to highly pathogenic avian H5N1 [132], and influenza B viruses infect domestic pigs and cause influenza-like symptoms and lung lesions [133]. The pathogenesis of coinfection between H1N1 influenza virus and swine Streptococcus suis serotype 2 was investigated in a swine model [134]. Due to the possession of receptors for both avian and human influenza viruses, pigs are potential intermediate hosts for the adaptation of avian influenza viruses to humans [135]. In fact, using a swine model, Rajao et al. recently detected two novel reassortant, human-like H3N2 and H3N1 influenza viruses [136].

Numerous studies have shown that swine are suitable for evaluating influenza vaccine efficacy. A previous study reported similarities in the innate and adaptive immune responses of humans and swine, including elevation of inflammatory cytokines (IL-6, IFN-γ, IFN-α, TNF-α, and IL-10) after IAV infection [131]. HA-specific neutralizing antibodies were demonstrated to correlate with protection against influenza infections in swine, which is also seen in humans [131]. As in humans, an important role of CD4^+^ and/or CD8^+^ T cell responses for influenza virus clearance was also observed in infected pigs [131]. Recently, McNee et al. established the swine model for evaluating influenza monoclonal antibody treatment. Treatment using monoclonal antibodies significantly reduced viral load and pathology in this model [137]. Studies to understand and evaluate vaccine-associated enhanced respiratory disease (VAERD), which results in the intensification of clinical signs and the development of more severe lung lesions, have also been performed using the swine model [108]. 

Although pigs are suitable animal models for the study of influenza, the high cost, difficulties in animal handling, housing, and waste management make them less favorable [126]. Recently, the microminipig animal model was established to overcome the limitations posed by the large size of regular pigs in influenza virus studies. There was no significant difference in genomic sequence, receptor distribution, and susceptibility to influenza A viruses between microminipigs and regular-sized pigs, indicating that this model could be a good alternative model for influenza studies [138].

## 8. Feline and Canine Models

Canine and feline species were not considered natural hosts of influenza virus until an H3N8 equine influenza virus was isolated from dogs in 1999. This was followed by the identification of an H3N2 avian influenza virus in domestic dog populations in Asia and in the United States in 2004, and an outbreak of H7N2 among cats in the United States in 2016 [139]. 

The susceptibility of dogs to avian influenza without prior adaptation was attributed to the abundance of α2, 3-linked sialic acid receptors in their lower respiratory tract [140]. Further, H5N1 infection of dogs is possible due to their close contact with wild and domestic birds as well as humans. Following experimental infection, clinical manifestation in dogs was characterized by the development of conjunctivitis and fever within 2 days, which resolved with no other adverse events by day 7. While dogs have been found to be susceptible to experimental infection, thus far, they have not been found to be capable of transmitting the virus to other mammals [141]. However, continued surveillance for canine influenza infections is imperative due to the high possibility that the virus can adapt to dogs and could transmit to humans while retaining its virulence. Given the increasing number of companion animals (dogs and cats) worldwide, their close contact with humans is a potential risk for zoonotic transmission of infectious viral diseases like influenza. In this sense, dogs may be considered more useful as a sentinel for human disease than as a model of human influenza [142].

In 2004, a study of highly pathogenic avian influenza H5N1 using a feline animal model showed severe and fatal disease manifestation as well as transmission of the virus via direct contact between infected and naive animals [143]. The virulence of H5N1 in cats was assessed through intratracheal infection and ingestion of virus-infected chicks. Clinical signs such as fever, conjunctivitis, lethargy, and labored breathing were observed in infected animals. After inoculation, the virus systematically spread and was isolated from the respiratory tract, digestive tract, liver, kidney, heart, brain, and lymph nodes. Moreover, the virus was found to be shed in the respiratory tract and also in the stool [143]. Although cats were reported to have susceptibility to IAV infection and there has been some work toward utilizing a feline influenza model for evaluating pathogenesis of H5N1 infections, few (if any) studies have been published which describe the suitability of felines for efficacy studies of influenza vaccines and antivirals [141,144].

## 9. Non-Human Primate Model

The close genetic relationship of non-human primates (NHPs) to humans, coupled with anatomic, physiologic, and immune feature similarities, makes this animal model invaluable for studying seasonal and highly pathogenic influenza viruses [145]. There are several NHP species that have been used for influenza studies, including rhesus (*Macaca mulatta*), cynomolgus (*Macaca fascicularis*), and pig-tailed (*Macaca nemestrina*) macaques, African green monkeys (*Chlorocebus aethiops*), New World NHPs, and common marmosets (*Callithrix jacchus*) [146]. While rhesus macaques have been selected for pathogenic studies and the evaluation of therapeutic and prophylactic strategies [147], the pig-tailed macaque is recommended for transcriptional studies of influenza virus [148]. Cynomolgus macaques have recently been used for various influenza studies [149,150,151].

NHPs were found to be susceptible to many avian and human influenza A isolates without prior adaptation. Clinical symptoms can range from asymptomatic to mild infections, which include conjunctivitis, listlessness, anorexia, and nasal discharge, all consistent with human clinical manifestations [145,147,152]. A primate model was also used to study the pathogenesis of H5N1 influenza virus infection [153]. Recently, Fukuyama et al. used aged cynomolgus macaques to study the severity of H7N9 influenza infection and the mechanism underlying the increased susceptibility of the elderly [154]. Nonetheless, NHPs are rarely used as models for influenza virus transmission. In 2013, Moncla et al. first demonstrated that human influenza virus isolates could infect and transmit between common marmosets, suggesting that this model could be used for studying influenza virus transmission [155].

NHPs are an overlooked animal model for evaluation of the immunogenicity and efficacy of novel influenza vaccine candidates and treatments. Cross-reactive T cells have been shown to be involved in the rapid viral clearance and heterologous protection of NHPs against pandemic H1N1 influenza virus [156,157]. Suzuki and colleagues employed cynomolgus macaques to investigate the efficacy of a cap-dependent endonuclease inhibitor and neuraminidase inhibitors on protection against high pathogenic avian H7N9 influenza virus [149]. A study by Darricarrère et al. investigated the ability of an HA stem-based influenza vaccine to induce broadly cross-reactive neutralizing antibodies against H1 and H3 using an NHP model [158]. The potential effects of quadrivalent influenza nanoparticle vaccines was also investigated in NHPs [159]. Furthermore, Koutsakos et al. recently developed flow cytometry-based assays to detect influenza-specific B and T cells in cynomolgus macaques. The use of these assays could be helpful for studying immune responses to influenza viruses in this model [151].

To date, NHPs have been used in many laboratories in order to study influenza virus pathogenesis and evaluate influenza vaccine and antiviral drug efficacies. However, ethical considerations, prohibitive costs, complicated husbandry requirements, and the need for extremely experienced personnel make this model less accessible for influenza research.

## 10. Conclusions

Animal models play a crucial role in influenza research for many reasons, e.g., study of influenza pathogenicity and transmission, evaluation of the impact of viral-host interactions, and investigation of vaccine and antiviral drug efficacy. Due to convenience, mice are the most commonly used model; however, ferrets are the best choice for studies of influenza transmission as well as vaccine and therapeutic efficacy. Other animal models, including guinea pigs, swine, felines, canines, and NHPs, also contribute to our understanding of influenza virus pathogenesis. There is no single animal model that perfectly recapitulates influenza disease in humans. Thus, each animal model has distinct advantages and disadvantages (Table 3), which should be taken into consideration when selecting the appropriate animal model(s) for each specific research question.

## Figures and Tables

**Table 1 viruses-13-01011-t001:** Mouse strains used in influenza virus research.

Mouse Strain		Research Application	Reference
Wild-type mice	C57BL/6	Pathogenesis, vaccine efficacy, and antiviral drugs	
	BALB/c	Pathogenesis, vaccine efficacy, and antiviral drugs	
	Infant C57BL/6	Transmission	[21]
Knockout/ deficient/ transgenic mice	DBA/2J	Pathogenesis and vaccine efficacy	[8,9]
	RAG1^−/−^	Role of B and T cells	[25]
	RAG2^−/−^	Role of B and T cells	[26]
	SCID	Role of B, T, and natural killer cells	[27]
	CCR2^−/−^	Role of monocytes	[28]
	B cell^−/−^	Role of B cell	[29,30,31,32]
	CD8^−/−^	Role of CD8 T cell	[30,33]
	CD4^−/−^	Role of CD4 T cell	[30,31,32,34]
	IFNR^−/−^	IFN signaling pathway	[35,36]
	B6-Mx1^−/−^ B6-Mx1^r/r^ SPRET/Ei	Role of Mx1 gene in virus resistance	[37,38]
	Tmprss2^−/−^	Pathogenesis	[10,11,12,13]
	IFITM3^−/−^	Influenza-induced cardiac pathogenesis	[45]
Humanized mice	DRAG	Generation of cross-reactive, human anti-influenza monoclonal antibodies and study of viral transmission	[22,39]
	DRAGA		
	Rag2^−/−^γc^−/−^	Pathogenesis and antiviral drug	[40]
	HLA-A2	Vaccine efficacy	[41]
	NOD/SCID β2m^−/−^	Vaccine efficacy	[42]
	NOD/SCID/Jak3^−/−^ (NOJ)	Vaccine efficacy	[43]
	NOD/Shi-scid *IL2rγ^null^*	Acute toxicity of an influenza vaccine	[44]

**Table 3 viruses-13-01011-t003:** The advantages and disadvantages of animal models in influenza research.

Animal Model	Advantages	Disadvantages
Mice	Easily manipulated genome Ease of handling and ease of husbandry Low cost including housing, maintenance and reproduction Decreased variability of inbred mouse strains Used for pathogenesis study, vaccine and antiviral drug efficacy test Availability of virological and immunological reagents	Mice are not naturally infected with influenza Difference in disease manifestation and pathogenesis Not suitable for transmission studies Difference between murine and human immune response
Ferret	Anatomically and physiologically comparable to human respiratory tract Comparable lung pathology to humans Suitable for pathogenesis, transmission and vaccine efficacy studies	Lack of ferret specific immunological reagents Cost of the animal and handling
Guinea pig	Naturally susceptible to various influenza strain without prior adaptation	Few to no clinical signs Lack of reagents
Hamster	Ease of handling Low maintenance cost Susceptibility to influenza virus Used in transmission and vaccine efficacy studies	No clinical signs
Chicken	Low costs, high availability, with medium availability of species-specific reagents and could also be genetically modified Used for influenza virus pathogenesis, transmission and vaccine efficacy studies	Not susceptible to human influenza virus strains
Swine	Abundance of both avian and mammalian influenza receptors in the respiratory tract Marked similarity in genome sequences, anatomy and physiology with humans Used in transmission studies	High cost Difficulty in handling
Feline and Canine	Abundance of α2, 3-linked sialic acid receptors in their lower respiratory tract Used in surveillance studies	Not a natural host of influenza virus not capable of transmission
Non-human primates	Close genetic relationship with humans (anatomic, physiological, immune feature similarities) Susceptible to many human and avian influenza virus Comparable lung pathology to humans Used in pathogenesis and vaccine efficacy studies	Ethical considerations High cost Complex husbandry requirements

## Data Availability

Not applicable.

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
