# Peer review of "Animal Models for Influenza Research: Strengths and Weaknesses"

_viruses, 2021, doi:10.3390/v13061011_

Round 1

Reviewer 1 Report

The review entitled ‘Animal models for influenza research: strengths and weakness’ is a very clear and well written manuscript. The authors do a comprehensive summary of the literature that imparts a lot of knowledge regarding the available animal models used for influenza virus research. The pros and cons are outlined for each are outlined and the most recent studies are nicely summarized in a table.

Ref #3 refers to a paper describing immune-evasion mechanisms. The statement that is being made refers to morbidity and mortality. Please cite a more appropriate reference or book chapter.

Please specify the subtype of each pandemic on lines 32-33.

Line 43. Please specify that pandemics do not arise from mutation but reassortment.

It would be helpful to the reader if the authors pointed out specifically whether a reference is examining contact or airborne transmission as well as whether the influenza strain used in the study was host-adapted or of clinical importance to humans.

I feel that at times statements are made about the animal model that was used in a particular study but not the outcome of that study. For example, the use of the microminipig as a model system – it would be interesting to tell the reader whether or not that was a good model or not.

I think that a figure outlining the pros and cons of all animal models would be quite informative to summarize this review.

Reviewer 2 Report

* Reviewing comments for the authors:

Animal models play a crucial role in influenza research, including for the investigations of influenza pathogenesis, transmission, evaluation of the impact of viral-host interactions, and the efficacy of vaccine and antiviral drugs. In this manuscript, Nguyen et al. have a broad overview of suitable animal models applied for influenza research. Overall, the authors gave a comprehensive summary of current applicable animal models for influenza research, including pathogenesis, transmission, efficacy evaluations of vaccines and therapeutic agents. The authors made clear descriptions and summarizations for several critical animal models which are commonly applicable in influenza research, pointed out the strength and weaknesses of each animal. In addition, the authors also proposed some research gaps and the limitations of each animal. However, it still exists some paragraphs that need to be further clarified or included more detailed descriptions.

1). Hamster models are also a recent emerging animal model applied in influenza research, including the pathogenesis and transmission…etc, whereas it missed in this manuscript, please add this animal model in the context.

2). Chicken is the critical species for avian influenza research as well, whereas this animal model was not included in the manuscript, please replenish it.

3). Throughout the manuscript, the authors described the strength and weakness of each animal model, respectively. If the authors could further make a summary table to sum up the application areas for each animal, such as transmission, pathogenesis, the evaluation of vaccine/therapeutic agents, and describe the current research gaps for each animal, it will help the readers have a clearer understanding for the advantages and limitations for each animal model. This information will allow readers to have clear reference and select an appropriate animal model(s) based on their specific scientific questions.
